# Eosinophilic Fasciitis: Current and Remaining Challenges

**DOI:** 10.3390/ijms24031982

**Published:** 2023-01-19

**Authors:** Diana Mazilu, Laura Alina Boltașiu (Tătaru), Denise-Ani Mardale, Maria Silviana Bijă, Sermina Ismail, Violeta Zanfir, Florentina Negoi, Andra Rodica Balanescu

**Affiliations:** 1“Sfanta Maria” Clinical Hospital, Ion Mihalache blvd, 37-39, 011172 Bucharest, Romania; 2“Carol Davila” University of Medicine, Dionisie Lupu Street, nr 37, 020021 Bucharest, Romania

**Keywords:** eosinophilic fasciitis, Shulman, systemic sclerosis

## Abstract

Eosinophilic fasciitis (EF), defined as diffuse fasciitis with eosinophilia by Shulman in 1974, is a disease with unknown etiology and whose pathogenesis is still being researched. The diagnosis is based on the clinical aspects (skin induration with an “orange peel” appearance), the lab results (eosinophilia, increased inflammatory markers), the skin biopsy with the pathognomonic histopathological result, as well as the typical MRI changes. The treatment includes glucocorticoids and immunosuppressive drugs. Due to severe refractory cases, the treatment remains a challenge. EF is still a disease with potential for further research.

## 1. Introduction

Eosinophilic fasciitis (EF) is a rare disease described by the presence of pitting edema and erythema on limbs or trunk and later by collagenous thickening of the subcutaneous fascia, hypergammaglobulinemia, eosinophilia in the peripheral blood, and diffuse fasciitis on the histopathologic exam.

In 1974, Shulman proposed a new disease concept called “diffuse fasciitis with eosinophilia” based on two cases: both patients with bilateral, symmetrical, diffuse, scleroderma-like skin induration on the limbs, joint contracture, no signs of Raynaud phenomenon or internal organ lesions, and with good response to oral glucocorticoid therapy.

In 1975, Rodnan proposed the name “eosinophilic fasciitis”, the most accepted term, even if, in certain stages of the disease, specific disease characteristics, such as peripheral eosinophilia and eosinophilic infiltration in the hypertrophied fascia, may be absent.

EF etiology is unknown, and pathogenesis is poorly understood. Thus, its treatment remains challenging. The most well-known triggering factor is sustained intense physical exercise. Other extrinsic factors, such as *Borrelia burgdorferi* infection or exposure to certain drugs, have also been mentioned. EF has been described as being associated with pathologies of the onco-hematological spectrum as well as with autoimmune ones. The exclusion of these primary causes is essential. Treatment in EF is by high-dose glucocorticoid therapy and immunosuppressive drugs, such as methotrexate, hydroxychloroquine, mycophenolate mofetil, and others. Some patients need surgical interventions for complications like joint contractures or carpal tunnel syndrome. This article is a review of the lately published data regarding EF etiology, clinical presentation, diagnosis, and new treatment options.

## 2. Etiology

Most cases of EF are considered idiopathic. However, a few possible triggers or associated factors could be: strenuous exercise [1];some of the autoimmune diseases (systemic lupus erythematosus, Sjögren syndrome, primary biliary cirrhosis, thyroid disease);exposure to certain medications (such as statins, ramipril, heparin, pembrolizumab, immune checkpoint inhibitors, and anti-tumor necrosis factor agents) [2,3,4,5,6];hematologic disorders (such as immune-mediated anemia or thrombocytopenia, pancytopenia, aplastic anemia, pure red cell aplasia, Hodgkin lymphoma, myelomonocytic leukemia, chronic lymphocytic leukemia, other leukemias and lymphomas, multiple myeloma, and other myeloproliferative disorders); cancers [melanoma, lung]initiation of hemodialysis [7,8,9];infection with *Borrelia burgdorferi* [10];use of adulterated rapeseed oil (epidemic outbreak in Spain in 1980s)radiotherapygraft-versus-host disease [11]

Several cases of patients who had eosinophilic fasciitis as an adverse reaction after treatment with nivolumab were reported [2,12,13,14,15]. Skin lesions in EF can be caused by several factors. Tissue inhibitor metalloproteinases (TIMPs) regulate the deposition of extracellular matrix by inactivating matrix metalloproteinases. Elevated levels of TIMPs have been reported in patients with EF, leading to an increased amount of extracellular matrix in these patients. Similarly, CD8 T cells are responsible for the production of TIMPs and/or stimulating other cells to produce TIMPs. Nivolumab’s overactivation of T cells could then be an additional culprit of the fibrosis seen in EF. Two other cases of EF, treated with Pembrolizumab, were reported secondary to this treatment [6,16].

## 3. The Pathogenesis of EF

EF is considered by some authors to be part of the spectrum of localized scleroderma, as is morphea. About 29–40% of patients with EF simultaneously present morphea. The pathological mechanisms involved in these diseases remain to be elucidated [17,18,19].

An autoimmune mechanism was considered to be essential, based on clinical aspects, laboratory results, and good response to glucocorticoid therapy. It was observed that in patients with EF, dermal fibroblasts show a greater expression of fibronectin and type I collagen. On the other hand, fibrosis is also triggered by the excessive production of the inhibitor of the extracellular matrix degradation enzyme metalloproteinase-1 (MMP-1, collagenase), namely the tissue inhibitor of metalloproteinase-1 (TIMP-1) [17].

Eosinophilia seems to play an important role in the pathogenesis of EF through elevated levels of eosinophilic cationic protein and serum interleukin-5 (IL-5) and increased eosinophilic migration capacity [20]. The increased level of histamine in the plasma shows the involvement of mast cells in the pathological process. At the same time, increased production of IL-2, interferon-γ (IFN-γ), and leukemia inhibitory factor (LIF) by peripheral blood mononuclear cells, overexpression of CD40 ligands and elevated superoxide dismutase (SOD) levels were objectivized [18,20]. The level of Th+ cells is increased in this condition [21].

On the other hand, an increase in the expression of transforming growth factor-β1 mRNA in fascia-derived fibroblasts and an increase in the expression of connective tissue growth factor genes in fibroblasts from the fascia of the affected areas were observed. Thus, these fibrosis-related cytokines are also involved in the physiopathology of the disease [18,20].

## 4. Histology

The biopsy will highlight edema and inflammatory infiltrate with lymphocytes, plasmocytes, histiocytes, and most often eosinophils in the deep fascia and lower subcutis. Later, the fascia becomes thickened and sclerotic, with the disappearance of the inflammatory infiltrate. After the degranulation of eosinophils in the fascia (which contains cytokines, chemokines, and growth factor), proteins are released and accumulated (such as cationic granule proteins ECP, eosinophil-derived neurotoxin EDN, eosinophil peroxidase EPX, and major basic protein), with toxic and fibrotic potential. Histamine, the degranulation product of mastocytes, is observed both in the affected tissues and in the circulating levels of the patients [18,20].

These histological changes are also found at the level of the subjacent muscle, being interested in both epi-, peri- and endomysium, but also muscle fibers. Comparative studies were carried out between the biopsy results obtained from other idiopathic inflammatory myopathies (such as poli- and dermatomyositis) and that from EF. The pathological process involves not only the fascia but also other muscle structures in different degrees.

## 5. Clinic

In 50% of cases, the onset of the disease is sudden [22]. Most of the time, a careful anamnesis reveals a recent history of intense physical exercises before the appearance of the first signs or symptoms of the disea [1]. The most frequent clinical features include skin changes, arthralgia and/or arthritis, myalgia and/or myositis, and rarely neuropathies and serositis [22].

The skin damage is often bilateral, and the changes go through several stages. Initially, the skin acquires a non-pitting edema appearance on the full-circumference of the distal limbs (forearms and lower legs). In the early stages, redness and local pain can be associated. Fever and fatigue were observed in many patients [22]. Later, the edema is replaced by symmetrical induration with puckering that gives the skin the texture of “orange peel”. A typical change is a linear depression that follows the path of the vessels in the affected area, known as the “groove sign”.

The territories of interest for the appearance of skin lesions are as follows: extremities, trunk, and neck. The skin of the hands and feet is generally spared, and the scleroderma of the fingers (sclerodactyly), a distinctive sign in systemic scleroderma, is absent, which helps the differential diagnosis. At the same time, the irregular, woody surface of the “orange peel”, compared to the smooth, shiny skin found in systemic scleroderma, pleads for the EF.

The musculoskeletal manifestations are the main reason the patients are referred to a rheumatologist. Inflammatory arthritis is present in less than half the patients diagnosed with EF and is usually located near the indurated fascia, which has important consequences in the degree of mobility of the joints. The muscle damage is expressed as myalgia and muscle weakness [18].

Some patients may develop neurological manifestations. Secondary carpal tunnel syndrome is often seen due to local compression on the median nerve in the affected joints. Visceral involvement is rare. There are a few reported cases of pulmonary, pleural, renal involvement, and pericarditis [17].

## 6. Investigations

Patients’ laboratory tests usually show transitory peripheral eosinophilia, not correlated with the severity of the disease, increased acute phase reactants (erythrocyte sedimentation rate and C-reactive protein), as well as a polyclonal hypergammaglobulinemia [17]. Immunoelectrophoresis is essential for the exclusion of the previously mentioned hematological diseases that may trigger EF. There is data suggesting that eosinophilia may be transient and tissue eosinophilia may disappear before achieving normal blood eosinophil levels. This finding supports the need for further investigations (such as biopsy and/or MRI) in patients with high clinical suspicion of EF [23]. The degree of activity of the disease can be appreciated by the increased values of serum aldolase, modified in many of the patients with EF, as well as the level of serum type III procollagen peptide (PIIIP), a marker that reflects disease activity and may be useful for monitoring [22].

Specific antibodies are usually absent. Anti-centromere, anti-topoisomerase 1, and anti-RNA polyisomerase III antibodies can be positive in 15–20% of EF patients. To exclude eosinophilic granulomatosis with polyangiitis, the assessment of anti-neutrophil cytoplasmic antibodies (ANCA) is recommended [17]. Antinuclear antibodies and rheumatoid factor were identified in 10% of EF patients [22].

The most important investigation for the positive diagnosis remains the biopsy which should include deep muscular fascia.

When the biopsy is not conclusive or cannot be performed, nuclear magnetic resonance (MRI) of the affected areas can be useful in highlighting the fascial inflammation. Fascial inflammation on MRI is confirmed by the increased T2 signal in the subcutaneous and deep fascia and the enhancement of the structures on fat-suppressed T1 images after gadolinium administration. Other imaging investigations, such as ultrasound and 18F-fluorodeoxyglucose positron emission tomography/computed tomography (FDG-PET/CT), may be chosen if MRI is not possible or is contraindicated [24].

## 7. Diagnostic Criteria and Disease Severity

Diagnostic criteria for EF represented an important research topic for many years. The last classification criteria were published in the *Journal of Dermatology* in 2018 [25,26,27,28]. Even though this classification criteria need further validation through international collaboration, it is an easy approach for the diagnosis of EF in clinical daily practice. The classification criteria (see Table 1) include one major criterion—symmetrical plate-like sclerotic lesions located on the limbs, the absence of Raynaud’s phenomenon, and the exclusion of systemic sclerosis. There are two minor criteria: the histologic aspect of the skin biopsy that incorporated the fascia and the specific changes seen on MRI [25,26,27,28].

A patient meets the classification criteria if the patient has the major criterion and one of the minor criteria or the major criterion and two of the minor criteria.

Regarding the EF severity, a classification score was proposed (see Table 2) [25]. The following items are scored with one point each: joint contracture (upper limbs), joint contracture (lower limbs), limited movement (upper limbs), limited movement (lower limbs), expansion, and worsening of skin rash (progression of symptoms). A total score of 2 or more points is classified as severe [25,26,27,28].

This severity score is useful in clinical practice to assess treatment response. Besides all the available immunosuppressive drugs, the lack of therapeutic response remains an important issue contributing to disease severity. The age of disease onset is crucial. Unfortunately, juvenile EF may present with systemic involvement and long-term disabling outcome [29]. Nonetheless, EF may present as a paraneoplastic syndrome or as an adverse event to chemotherapy [2,3,6]. In these cases, the patient’s prognosis also depends on the underlying disease.

## 8. Differential Diagnosis

Differential diagnosis includes other conditions that present with skin induration or tissue fibrosis.

The differential diagnosis should begin with other pathologies that are part of the same spectrum of localized sclerodermas, such as morphea (linear and diffuse) or pansclerotic scleroderma. These conditions do not associate with eosinophilia and have a slowly progressive course.

**Systemic scleroderma** and EF are two distinct entities. The Raynaud phenomenon is absent in EF patients, but it is the primary clinical manifestation in approximately 95% of patients with systemic scleroderma with diffuse or localized skin damage. Performing capillaroscopy in these patients provides information about the capillary architecture, which is normal in patients with EF, compared to a disorganized capillary architecture, dilated capillaries, microhemorrhages, avascular areas, that is found in most patients diagnosed with scleroderma. The normal capillaroscopic pattern does not exclude systemic sclerosis, and neither confirms EF. Another distinctive aspect is that in EF, digital pitting scars are absent, and also, the skin damage does not involve the hands, feet, and face. A very important criterion, which differentiates these two diseases, is internal organ damage (e.g., pulmonary fibrosis, renal crisis, pulmonary hypertension), while EF is usually limited only to skin involvement. And last but not least, the presence of the specific serology in scleroderma outlines the differential diagnosis.

Scleroderma-like entities have been described in the literature, which must be differentiated from EF. For example, **nephrogenic systemic fibrosis** is found in patients with advanced kidney disease (dialysis-dependent or with a glomerular filtration rate <15 mL/min), with or without the association of gadolinium administration. Compared to EF, nephrogenic systemic fibrosis affects distinct areas (hands and feet); eosinophilia is absent, and a distinct histopathologic pattern is present [30].

**Scleromyxedema** is a condition highlighted by the deposition of amorphous mucinous material in the dermis, which causes thickening of the skin, but also by the appearance of yellow–red waxy papules. It is frequently associated with monoclonal gammopathy [31].

Patients with **scleredema** present with diffuse induration of the skin, the absence of autoantibodies, and the absence of signs of inflammation on skin biopsy. This disease is most often found in patients with diabetes, usually treated with insulin, but it has also been described as associated with monoclonal gammopathy [32].

The diseases that associate with eosinophilia and skin thickening should be ruled out. In this category, an important differential diagnosis is **eosinophilia–myalgia syndrome** (EMS). The consumption of supplements based on L-tryptophan or 5-hydroxytryptophan is cited as an etiology of EMS. The dominant clinical sign is severe myalgia. Myalgia from EF is much less common and much milder. Regarding the skin damage from the two pathologies, it is similar, but in EMS, there is also visceral involvement, such as pneumonitis and neuropathy, whereas these are not found in EF [33].

Another entity that associates eosinophilia and skin induration is **toxic oil syndrome**. An epidemic outbreak was described in the literature in the 1980s in Spain, caused by adulterated rapeseed oil. It is characterized by dyspnea, myalgias, arthralgias, edema, and skin induration in the limbs, similar to that seen in scleroderma, livedo reticularis, joint contractures, and neuropathy, as well as eosinophilia, increased serum creatinine kinase, and pulmonary infiltrates. **Graft-versus-host disease** is also associated with skin induration and fibrosis. Sclerotic skin changes may appear in any areas of previously normal skin or in the areas of resolving lichen planus-like lesions, particularly on the medial arms and thighs. Deep sclerosis with fascial involvement and typical “groove sign” appears in the chronic period of this disease.

Summarizing, a scheme may be useful to ease the diagnosis process in clinical practice (see Table 3).

## 9. Treatment and New Potential Targets for EF

The therapeutic approach in EF is currently unclear; there are also no randomized studies regarding therapy in EF. A part of our personal medical experience was published recently [24]. The initial treatment consists of a dose of 1 mg/kgc per day of Prednisone with subsequent tapering [17]. The rapid resolution of the eosinophilia and the normalization of the ESR values are objective, but the softening of the affected skin may take weeks to months. Higher doses of glucocorticoids may be considered if eosinophilia or signs and symptoms of EF persist. Patients on high-dose glucocorticoid treatment are at risk of osteoporosis and opportunistic infections; thus, they are potential candidates for antiresorptive therapy to prevent bone loss as well as prophylaxis against *Pneumocystis jirovecii*, which is often indicated.

Other immunosuppressive and immunomodulatory agents are considered to obtain a therapeutic response or to spare the use of glucocorticoids, as well as in patients unresponsive to 1.5 mg/kg/day Prednisone administered for three months [17]. Methotrexate is chosen as subsequent therapy. It can be administered in a dose of 15 mg to 25 mg per week. Once remission is achieved, the duration of therapy can be maintained between four and six months, then stopped. Other alternatives would be mycophenolate mofetil or hydroxychloroquine, but there are limited data on their therapeutic effect [34].

In the case of patients with the refractory disease to conventional therapies, other therapies based on case series or case reports can be considered, such as tocilizumab [35], baricitinib [36], sulfasalazine, azathioprine, infliximab, rituximab, intravenous immunoglobulins, dapsone, phototherapy with ultraviolet A (UVA) 1, and psoralen plus photochemotherapy UVA (PUVA) [37]. Successful treatment was achieved in patients treated earlier in the course of the disease with cyclosporine A in combination with glucocorticoids or other immunosuppressive therapies. The duration of treatment in patients without severity criteria can last for around two years [17,38,39].

In refractory cases, new drugs have been tried, such as sirolimus—an inhibitor of rapamycin kinase. There are a few case reports showing its efficiency in patients non-responding to glucocorticoids and methotrexate. It may be an alternative therapy [40].

There are no clinical studies regarding the use of extracorporeal photopheresis (ECP) in the treatment of EF, but there are successful medical case reports published. It may be useful for patients with steroid-resistant EF or for those with inadequate response to other treatment regimens [41].

Concomitant with immunosuppressive treatment, physical therapy is essential for maintaining joint mobility and decreasing contractures [42]. In severe cases, surgical interventions can be used in the case of joint contractures in association with glucocorticoid therapy. Additionally, surgical fasciectomy or surgical treatment for carpal tunnel syndrome can be performed in patients who do not respond to conventional therapy.

Some other cases of refractory EF in conventional therapies have been cited. For these patients, reslizumab, an infusion-based humanized monoclonal antibody with anti-IL-5 activity, was tried. It was efficient for symptom resolution and cortisone cessation [43,44]. A clinical study using mepolizumab, another anti-IL-5 agent, is estimated to start soon [45].

## 10. Discussion

Possible causal factors that could trigger this disease are cited in the literature. The most well-known triggering factor is sustained intense physical exercise. Other extrinsic factors, such as *Borrelia burgdorferi* infection or exposure to certain drugs, have also been mentioned. EF has been described as being associated with pathologies of the onco-hematological spectrum as well as with autoimmune ones. The exclusion of these primary causes is essential.

The pathological process is currently incompletely known. However, the theory of the involvement of an autoimmune mechanism is supported by the clinical and laboratory aspects of the disease and also by the good response to corticosteroids. The diagnosis is supported by the clinical characteristics, laboratory, histological, and imaging aspects.

EF is remarked by typical skin lesions, such as the “orange peel” appearance and the “groove sign”. The skin on the hands, feet, and face is spared, and EF does not cause visceral damage. In addition, Raynaud’s phenomenon is absent. Inflammatory arthritis is present in less than half of patients diagnosed with EF and is usually located near the indurated fascia. Among the symptoms encountered in EF is myalgia, but not always present and not very severe as is otherwise described in eosinophilia–myalgia syndrome [44]. The diagnosis of this disease is mainly based on the exclusion of other pathologies, such as localized sclerodermas (morphea, pansclerotic scleroderma), systemic scleroderma, and scleroderma-like entities (nephrogenic systemic fibrosis, scleromyxedema, scleredema), as well as with the group of diseases that associate eosinophilia and skin thickening (eosinophilia–myalgia syndrome, toxic oil syndrome), graft-versus-host disease. The gold standard procedure for the diagnosis is the skin biopsy which should include deep muscular fascia. In situations where skin biopsy cannot be performed or it is contraindicated, MRI remains an alternative that will highlight fascial inflammation [46].

Treatment in EF is initially represented by high-dose glucocorticoid therapy. Higher doses of glucocorticoids can be considered if the targeted therapeutic response has not been achieved, but with the risk of adverse effects to which the patient is subjected. The second line, in case of non-responsiveness to cortisone in high doses or in the desire to spare the use of glucocorticoids, is methotrexate. Alternatives to methotrexate would be hydroxychloroquine and mycophenolate mofetil. In the case of patients with the disease refractory to conventional therapies, other therapies based on case series or case reports can be considered. Finally, last but not least, surgical interventions can be considered in patients with joint contractures or carpal tunnel syndrome.

## 11. Conclusions

Possible causal factors for this disease and the differential diagnosis were discussed above. The current challenges are the appearance of EF as an adverse event to the new therapeutic drugs, those of the onco-hematological and autoimmune diseases. This may limit the therapeutical spectrum of approach in some diseases. The theory of the involvement of an autoimmune mechanism is supported by the clinical and laboratory aspects of the disease and also by the good response to corticosteroids—the first line of treatment. The second line, in case of non-responsiveness to cortisone in high doses or in the desire to spare the use of glucocorticoids, is methotrexate or some other alternatives, such as hydroxychloroquine and mycophenolate mofetil. In the case of patients with the disease refractory to conventional therapies, other new therapies based on case series or case reports can be considered. A clinical trial using mepolizumab is planned to start soon. Surgical interventions can be considered in patients with joint contractures or carpal tunnel syndrome. Even though EF is a rare disease, in clinical practice, it is a challenging disease, with patient variability in etiology and treatment response. Publishing new case presentations and starting clinical trials may be a step forward toward a new therapeutical approach and defining a treatment algorithm.

## Figures and Tables

**Table 1 ijms-24-01982-t001:** Diagnostic criteria for EF.

Major Criteria	Minor Criteria
Symmetrical plate-like sclerotic lesions present on the four limbs.	1. The histology of a skin biopsy that incorporated the fascia shows fibrosis of the tissue, with thickening of the fascia and cellular infiltration of eosinophils and monocytes.
	2. Thickening of the fascia is seen using imaging tests, such as magnetic resonance imaging (MRI)

**Table 2 ijms-24-01982-t002:** Severity classification of EF.

Joint Involvement	Points
Joint contracture (upper limbs)	1 point
Joint contracture (lower limbs)	1 point
Limited movement (upper limbs)	1 point
Limited movement (lower limbs)	1 point
Expansion and worsening of skin rash (progression of symptoms)	1 point
A total of 2 or more points is classified as severe.	

**Table 3 ijms-24-01982-t003:** A simplified approach to the diagnosis/differential diagnosis of EF.

Skin Changes (Erythema, Swelling, Induration) + Peripheral Eosinophilia
** *Exclude other disorders:* **
Mono skin lesion: - Localized scleroderma (morphea, linear and diffuse or pansclerotic scleroderma)
Diffuse skin lesions: - systemic sclerosis, - scleroderma-like disorders (nephrogenic systemic fibrosis, scleromyxedema, scleredema - eosinophilia–myalgia syndrome (associated with the use of L-tryptophan supplements) - toxic oil syndrome (rapeseed oil use) - graft-versus-host disease - other systemic diseases: malignancies, infections, autoimmune diseases
**No systemic involvement** ↓
** *Proceed to full thickness skin-to-muscle biopsy or MRI of the affected area* **

## Data Availability

Not applicable.

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
