# Peer review of "Eosinophilic Fasciitis: Current and Remaining Challenges"

_ijms, 2023, doi:10.3390/ijms24031982_

Round 1

Reviewer 1 Report (Previous Reviewer 2)

The authors significantly improved their manuscript from the previous version. However, I still have some points to help improve this MS.

- The introduction is very short and not informative. The author should give a broader overview and situation e.g. line 34, "EF etiology is unknown and pathogenesis is poorly understood [1,2]. Thus, its treatment remains challenging.". Authors should provide more pieces of evidence and describe more. 

- Discussion is needed on several topics e.g. clinic, investigation, etc. The authors should discuss the proposed topic and provide more examples.

- The topic of this MS is "Eosinophilic fasciitis: current and remaining challenges". Thus, the authors should highlight the points of the current and remaining challenges according to the title.

Author Response

Response to Reviewer 1

Dear reviewer,

Thank you very much for your remarks and suggestions!

Please find below the responses.

The authors significantly improved their manuscript from the previous version. However, I still have some points to help improve this MS.

Point 1:  The introduction is very short and not informative. The author should give a broader overview and situation e.g. line 34, "EF etiology is unknown and pathogenesis is poorly understood [1,2]. Thus, its treatment remains challenging.". Authors should provide more pieces of evidence and describe more. 

Comment: These changes were made in the introduction section:

“The most well-known triggering factor is sustained intense physical exercise. Other extrinsic factors such as Borrelia burgdorferi infection or exposure to certain drugs have also been mentioned. EF, has been described as being associated with pathologies of the onco-hematological spectrum as well as with the autoimmune ones. The exclusion of these primary causes is essential.”

“Treatment in EF is by high-dose glucocorticoid therapy and immunosuppressive drugs, such as methotrexate, hydroxychloroquine, mycophenolate mofetil and others. Some patients need surgical interventions for complications like joint contractures or carpal tunnel syndrome.”

“This article is a review of the lately published data regarding EF etiology, clinical presentation, diagnosis and new treatment options.”

Point 2 - Discussion is needed on several topics e.g. clinic, investigation, etc. The authors should discuss the proposed topic and provide more examples.

Comment:

Our review covers the information that is published so far in the literature. The clinical presentation, the investigations and the treatment are discussed in the special sections, and also in the discussions. Regarding the examples, we have mentioned an article with 2 case presentations with patients treated in our clinic. As it was suggested by previous reviewers, we did not provide details regarding the clinics and treatment, as the article should cover more of the histopathological aspects of the disease.

Point 3 - The topic of this MS is "Eosinophilic fasciitis: current and remaining challenges". Thus, the authors should highlight the points of the current and remaining challenges according to the title.

The conclusion was changed:

“Possible causal factors for this disease and the differential diagnosis were discussed above. The current challenges are the appearance of EF as an adverse event to the new therapeutic drugs, those of the onco-hematological and autoimmune diseases. This may limit the therapeutical spectrum of approach in some diseases. The theory of the involvement of an autoimmune mechanism is supported through the clinical and laboratory aspects of the disease and also by the good response to corticosteroids – the thirst line of treatment. The second line, in case of non-responsiveness to cortisone in high doses or in the desire to spare the use of glucocorticoids, is methotrexate, or some other alternatives such as hydroxychloroquine and mycophenolate mofetil .[63] In the case of patients with refractory disease to conventional therapies, other new therapies based on case series or case reports can be considered. A clinical trial using mepolizumab is planned to start soon. Surgical interventions can be considered in patients with joint contractures or carpal tunnel syndrome. Even though EF is a rare disease, in clinical practice it is a challenging disease, with patient’s variability in etiology and treatment response. Publishing new case presentations and starting clinical trials may be a step forward for new therapeutical approach and for defining a treatment algorithm.”  

Reviewer 2 Report (New Reviewer)

The manuscript presents a correct description of a rare rheumatic disease - eosinophilic fasciitis.

Nevertheless, I did not find much news about the disease.

I suggest presenting the purpose of the article in the introduction.

The scope of information in manuscript is thematically similar to textbooks on rheumatic diseases or, for example, the "UpToDate" website, which the authors quoted four times.

I suggest citing original articles instead of didactic websites.

The discussion repeats data from previous parts of the manuscript. I suggest re-examining this part. In my opinion, the discussion should present new, according to the authors, information about the disease in order to emphasize the novelty of the article and the "current and remaining challenges" contained in the title.

In conclusion, the presented work, in my opinion, may be of practical importance for physicians. However, a re-examination of all parts of the manuscript, emphasizing new data, would give it more value.

Author Response

Response to Reviewer 2

Dear reviewer,

Thank you very much for your remarks and suggestions!

Please find below the responses.

The manuscript presents a correct description of a rare rheumatic disease - eosinophilic fasciitis.

Nevertheless, I did not find much news about the disease.

Point 1 - I suggest presenting the purpose of the article in the introduction.

Comment:

In this review we covered all the data that was published recently regarding this disease. Indeed, there are very few news, and mainly these refer to EF as an adverse event to new therapeutic drugs, especially those used in oncology, and also some clinical case presentation in which immunosuppressive drugs were tried.

Information added:

“The most well-known triggering factor is sustained intense physical exercise. Other extrinsic factors such as Borrelia burgdorferi infection or exposure to certain drugs have also been mentioned. EF, has been described as being associated with pathologies of the onco-hematological spectrum as well as with the autoimmune ones. The exclusion of these primary causes is essential.”

“Treatment in EF is by high-dose glucocorticoid therapy and immunosuppressive drugs, such as methotrexate, hydroxychloroquine, mycophenolate mofetil and others. Some patients need surgical interventions for complications like joint contractures or carpal tunnel syndrome.”

“This article is a review of the lately published data regarding EF etiology, clinical presentation, diagnosis and new treatment options.”

Point 2 - The scope of information in manuscript is thematically similar to textbooks on rheumatic diseases or, for example, the "UpToDate" website, which the authors quoted four times.

I suggest citing original articles instead of didactic websites.

Comment:

The UpToDate information uses articles from 1978-1980, 2004-2005 regarding this topic. As it was suggested previously, we updated the bibliography and tried to avoid older articles. This is the main reason we used UpToDate as citation. Also, there was information without a direct bibliographic source.”

Point 3 - The discussion repeats data from previous parts of the manuscript. I suggest re-examining this part. In my opinion, the discussion should present new, according to the authors, information about the disease in order to emphasize the novelty of the article and the "current and remaining challenges" contained in the title.

Comment: We agree. We tried to make changes according to reviewer’s notes. Since we have been suggested to discuss these matters in this section, we have made some changes in the conclusion section.

“Possible causal factors for this disease and the differential diagnosis were discussed above. The current challenges are the appearance of EF as an adverse event to the new therapeutic drugs, those of the onco-hematological and autoimmune diseases. This may limit the therapeutical spectrum of approach in some diseases. The theory of the involvement of an autoimmune mechanism is supported through the clinical and laboratory aspects of the disease and also by the good response to corticosteroids – the thirst line of treatment. The second line, in case of non-responsiveness to cortisone in high doses or in the desire to spare the use of glucocorticoids, is methotrexate, or some other alternatives such as hydroxychloroquine and mycophenolate mofetil .[63] In the case of patients with refractory disease to conventional therapies, other new therapies based on case series or case reports can be considered. A clinical trial using mepolizumab is planned to start soon. Surgical interventions can be considered in patients with joint contractures or carpal tunnel syndrome. Even though EF is a rare disease, in clinical practice it is a challenging disease, with patient’s variability in etiology and treatment response. Publishing new case presentations and starting clinical trials may be a step forward for new therapeutical approach and for defining a treatment algorithm.  “

In conclusion, the presented work, in my opinion, may be of practical importance for physicians. However, a re-examination of all parts of the manuscript, emphasizing new data, would give it more value.

This manuscript is a resubmission of an earlier submission. The following is a list of the peer review reports and author responses from that submission.

Round 1

Reviewer 1 Report

First of all I would like to commend the authors for their review on such a rare disease. It was a rather “refreshing” experience reading this compact and clinically oriented article.

My comments regarding this manuscript are only minor, as follows:

-      Please consider adding a visual synthetic scheme to better summarize the diagnostic/differential diagnostic process;

-      Shall the authors have encountered any recent cases of EF, it would be very interesting to see a short mention of their experience with the respective cases (1-2 figures with images and short clinical and therapeutic comments; something like "Clinical vignete");

-       A “(see Table 1)” paragraph should be incorporated in the article's body (Row 139 or 137) as a link to Table 1;

-       Please revise the use of English in the following identified situations:

o   Row 29: “EF etiology it is unknown and” – please consider dropping the “it”;

o   Row 206-207: “ but the affected skin softens, this can last from weeks to months” – presuming you meant that the softening of the skin might take weeks to months, please rephrase accordingly;

Reviewer 2 Report

The manuscript by Boltașiu and colleagues reveals current and remaining challenges of Eosinophilic Fasciitis (EF), a rare disease, which is being researched, especially its etiology, pathogenicity, and treatments. However, I have some comments to be addressed.

Major comments

1.      Lines 80 - 83: “Comparative studies were carried out between the biopsy results obtained from other idiopathic inflammatory myopathies (such as poli- and dermatomyositis) and that from EF without being able to differentiate [29].” – please also briefly mention the results of this study

2.      Line 210: Please describe more about “Graft-versus-host disease”. Because it is considered one of possible triggering factors or factors associated with EF.

3.      Lines 203 - 230: Due to the treatment of EF remains challenging, the authors could provide more information on treatment section. Methylprednisolone, extracorporeal photopheresis (ECP), cyclosporine, sirolimus, and/or D-penicillamine could be included. According to some case reports, physical therapy is recommended using in combination with systematic treatments for joint contractures in EF patients. In addition, the risk factors for treatment resistance, including concurrent plaque morphea, pediatric age of onset, and underlying disease, could be mentioned in the manuscript.

Minor comments

1.      Line 10: “(skin induration with an “orange peel appearance)” – “(skin induration with an orange peel appearance)”

2.      Lines 31 - 45: The content of etiology section could be re-managed to be easily understand such as adding the bullet in front of each EF-associated factor. Additionally, “radiotherapy” and “graft-versus-host disease” might be also included as factors associated with EF.

3.      Lines 44 and 234: “Borrelia burgdorferi” should be italic.

4.      Lines 56 and 82: “FE” – “EF”

5.      Line 67: “… are involved in physiopathology of the disease [25,26]” – full stop was missed.

6.      Line 212: “Pneumocystis jirovecii” should be italic.

7.      In Table 2, the titles of each column should be added.

8.      The citation of each sentence or phrase should be included in the sentence or phrase. For example, “initiation of hemodialysis [11]; …” or “The level of Th+ cells is increased in this condition [24].”.

9.      Some grammatical errors are detected in this MS e.g. line 29 “EF etiology it is unknown and pathogenesis is poorly understood. [1,2] Thus, its treatment remains challenging.”

Reviewer 3 Report

The authors have written a review on eosinophilic fasciitis, the title implies a focus on current and future challenges. In my opinion, this review fails to report and analyse recent reports on this disease, and hence, also fails to discuss "current and future challanges". 

I have some specific comments that may be of value to the authors:

-The 2nd sentence of the abstract is unclear and can be rewritten.

-I suggest not referring to previous review-articles when discussing our current knowledge. Also, I suggest to include appropriate references when for example discussing the EF as caused by pharmaceutical therapies. Also, additional references are needed when referring to the anecdotal (?) association with undue physical exercise.

-What do you mean with statements like "Visceral involvement is rare. Serositis and renal involvement, on the other hand, are much more common"?

-In addition, you later write "..in EF it is limited only to the skin"

-The suggested diagnostic criteria should be discussed in reference to the fact that they are not validated, they were not the result of any international collaboration.

-I disagree with your conclusion that "most cases respond promptly to glucocorticoid therapy or known immunsuppressives". This statement is not clearly supported in the review.

-The authors have done a good job reading original articles from the previous century, but only very few of articles on EF published during the last 4 years are included.

-The statement on PIIINP fails to discuss that this biomarker is not clinically available. Instead the author say it is a "very usefull marker in monitoring"

-The association with severe hematological disorders can be discussed further

-Laboratory testing including serum-electrophoresis and (transitatory) eosinophilia levels can be discussed more in relation to diagnosis

Round 2

Reviewer 2 Report

-

Author Response

No comments.

Reviewer 3 Report

The authors have made several minor adjustments in reference to my previous review. My main concerns persist

-I lack a discussion on the topic in reference to the most recent publications in the field.

-Also a more extensive discussion on the diagnosis of the disease is needed. Sentences such as "a definitive diagnosis is made when..." when referring to unvalidated classification criteria is unfortunate.

-When describing systemic sclerosis, the authors should be aware that many patients with systemic sclerosis have normal capillaries.

Round 3

Reviewer 3 Report

The manuscript has been improved. At this point, I do no longer find that it contain misleading comments  or selection of reported data. 

Several of my previous concerns remain, most importantly that it fails to add new perspective on this disease for which our current knowledge is fragmentary and anecdotal. Even though the bibliography is updated, I lack an in-depth analysis on how our knowledge on this rare and potentially serious disease have evolved during the last years.